# MicroRNA-223 Dampens Pulmonary Inflammation during Pneumococcal Pneumonia

**DOI:** 10.3390/cells12060959

**Published:** 2023-03-21

**Authors:** Cengiz Goekeri, Peter Pennitz, Wibke Groenewald, Ulrike Behrendt, Holger Kirsten, Christian M. Zobel, Sarah Berger, Gitta A. Heinz, Mir-Farzin Mashreghi, Sandra-Maria Wienhold, Kristina Dietert, Anca Dorhoi, Achim D. Gruber, Markus Scholz, Gernot Rohde, Norbert Suttorp, Martin Witzenrath, Geraldine Nouailles

**Affiliations:** 1Department of Infectious Diseases, Respiratory Medicine and Critical Care, Charité—Universitätsmedizin Berlin, Corporate Member of Freie Universität Berlin and Humboldt-Universität zu Berlin, 10117 Berlin, Germany; 2Faculty of Medicine, Cyprus International University, 99040 Nicosia, Cyprus; 3Institute for Medical Informatics, Statistics, and Epidemiology, Universität Leipzig, 04107 Leipzig, Germany; 4Department of Internal Medicine, Bundeswehrkrankenhaus Berlin, 10115 Berlin, Germany; 5Therapeutic Gene Regulation, Deutsches Rheuma-Forschungszentrum Berlin (DRFZ), ein Institut der Leibniz-Gemeinschaft, 10117 Berlin, Germany; 6Berlin Institute of Health at Charité—Universitätsmedizin Berlin, BIH Center for Regenerative Therapies (BCRT), 13353 Berlin, Germany; 7Institute of Veterinary Pathology, Freie Universität Berlin, 14163 Berlin, Germany; 8Veterinary Centre for Resistance Research (TZR), Freie Universität Berlin, 14163 Berlin, Germany; 9Institute of Immunology, Friedrich-Loeffler-Institut, 17493 Greifswald-Insel Riems, Germany; 10Faculty of Mathematics and Natural Sciences, University of Greifswald, 17489 Greifswald, Germany; 11Department of Respiratory Medicine, Medical Clinic I, Goethe-Universität Frankfurt am Main, 60596 Frankfurt am Main, Germany; 12CAPNETZ STIFTUNG, 30625 Hannover, Germany; 13Biomedical Research in Endstage and Obstructive Lung Disease Hannover (BREATH), German Center for Lung Research (DZL), 30625 Hannover, Germany; 14German Center for Lung Research (DZL), 10117 Berlin, Germany

**Keywords:** microRNA-223, pneumonia, *Streptococcus pneumoniae*, neutrophils, inflammation

## Abstract

Community-acquired pneumonia remains a major contributor to global communicable disease-mediated mortality. Neutrophils play a leading role in trying to contain bacterial lung infection, but they also drive detrimental pulmonary inflammation, when dysregulated. Here we aimed at understanding the role of microRNA-223 in orchestrating pulmonary inflammation during pneumococcal pneumonia. Serum microRNA-223 was measured in patients with pneumococcal pneumonia and in healthy subjects. Pulmonary inflammation in wild-type and microRNA-223-knockout mice was assessed in terms of disease course, histopathology, cellular recruitment and evaluation of inflammatory protein and gene signatures following pneumococcal infection. Low levels of serum microRNA-223 correlated with increased disease severity in pneumococcal pneumonia patients. Prolonged neutrophilic influx into the lungs and alveolar spaces was detected in pneumococci-infected microRNA-223-knockout mice, possibly accounting for aggravated histopathology and acute lung injury. Expression of microRNA-223 in wild-type mice was induced by pneumococcal infection in a time-dependent manner in whole lungs and lung neutrophils. Single-cell transcriptome analyses of murine lungs revealed a unique profile of antimicrobial and cellular maturation genes that are dysregulated in neutrophils lacking microRNA-223. Taken together, low levels of microRNA-223 in human pneumonia patient serum were associated with increased disease severity, whilst its absence provoked dysregulation of the neutrophil transcriptome in murine pneumococcal pneumonia.

## 1. Introduction

Community-acquired pneumonia (CAP) manifests itself as a lower respiratory tract infection that commences as local inflammation triggered by pathogen infiltration into the lungs [1]. Lower respiratory tract infections were the leading cause of global communicable disease-related mortality in 2019 [2] and can be commonly provoked by pathogens including *Streptococcus pneumoniae*, one of the main causative bacterial agents of lower respiratory tract infections [3]. Potential complications of CAP include acute respiratory distress syndrome (ARDS) and sepsis, both of which are major contributors towards intensive care unit admission [4]. The pathophysiology of ARDS [5], triggered by various lower respiratory tract infections including coronavirus disease 2019 (COVID-19) [6], includes increased recruitment of leukocytes, pronounced pulmonary edema together with compromised alveolar-capillary membrane permeability [5]. Polymorphonuclear leukocyte (PMN) extravasation is a part of the innate immune response in which leukocytes travel out of the circulatory system towards the site of tissue damage or infection. Movement of leukocytes into alveolar spaces commences within hours of pneumococcal infection [7] enabling them to exert pronounced antimicrobial effector functions to aid bacterial clearance [8,9]. If PMN effector functions are uncontrolled or prolonged, however, detrimental host-mediated damage may ensue, including hypoxemia and impaired mitochondrial function [10]. The further elucidation, identification and understanding of host factors regulating PMNs during development and progression of pneumococcal pneumonia is of key interest in understanding the pathophysiology of the disease and eventually improving its treatment in the clinical setting. Additionally, identifying factors involved in the regulation of the molecular cascade of PMN induction could aid in the development of novel diagnostic and prognostic biomarkers for severe CAP.

MicroRNAs (miRNAs) are short, non-coding single-stranded RNA molecules which can post-transcriptionally regulate expression of their target genes. Upon induction of their transcription, a longer primary transcript—which is 3′ poly-adenylated and 5′ capped—is initially transcribed by RNA polymerase II in the nucleus, which then gets processed by RNase II endonuclease III Drosha/DiGeorge syndrome chromosomal region 8 (DGCR8) to release the precursor miRNA [11]. This precursor is then shuttled to the cytoplasm whereby it undergoes further processing by RNase III Dicer-1 to produce a double-stranded miRNA duplex [12]. The guide strand of the miRNA duplex can then associate with the RNA-induced silencing complex (RISC) and specifically bind to the complementary 3′ untranslated region (3′UTR) of the target mRNA, resulting in either target mRNA degradation or repression of its respective translation [13]. Previous studies have shown microRNA-223 (miR-223) to play a vital role in various inflammatory diseases in that it binds to specific target genes inhibiting production of inflammatory mediators or blocking inflammatory signaling pathways, resulting in protection from inflammation-induced tissue injury [14].

MicroRNA-223 has also been described in regulating myeloid progenitor proliferation and function and is the non-coding master regulator of PMN function and maturation [15], thus making it a relevant miRNA in the context of the cellular immune response to bacterial lung infections. Absence of miR-223 has been shown to be detrimental in chronic lung infections, leading to dramatically reduced survival of mice infected with *Mycobacterium tuberculosis* (*Mtb*); a phenotype linked to persistent PMN responses in the lungs [16]. However, the exact pathology caused by absence of miR-223, and the resulting degree of inflammation and disease progression during acute pulmonary inflammation remain to be elucidated.

In this study, we investigate the role of miR-223 in the context of acute lung inflammation by employing a murine model of pneumococcal pneumonia. We describe the PMN-driven exacerbation of pulmonary inflammation in mice lacking miR-223 in addition to the transcriptional repercussions exhibited by murine lung cells in its absence using single-cell RNA sequencing (scRNA-seq). We further describe the association of serum miR-223 with blood inflammation parameters and disease severity in pneumococcal pneumonia patients. 

## 2. Materials and Methods

### 2.1. Mice and Housing 

Eight to twelve weeks old female C57BL/6J mice were sourced from the Research Institutes for Experimental Medicine (FEM) Berlin (Berlin, Germany) and Charles River (Sulzfeld, Germany) and age matched miR-223^−/−^ mice (kindly provided by Prof. Dr. Stefan H.E. Kaufmann) were kept under specific pathogen free conditions. All animal experiments were conducted in compliance with the Federation of European Laboratory Animal Science Association (FELASA) guidelines and approval of institutional (Charité—Universitätsmedizin Berlin, Berlin, Germany) and governmental (State Office for Health and Social Affairs (LAGeSo) Berlin, Berlin, Germany) authorities (G0266/11, G0139/14, T0078/14).

### 2.2. Intranasal Infection, Monitoring of Mice, and Anesthesia

For infection, baseline physiological parameters of mice (body weight, rectal body temperature) were recorded under sterile conditions. Animals were then given an intraperitoneal injection of a mixture of ketamine (93.75 mg/kg body weight; CP-Pharma, Burgdorf, Germany) and xylazine (15 mg/kg body weight; CP-Pharma, Burgdorf, Germany) diluted in sterile isotonic saline (B. Braun, Melsungen, Germany). Adequate depth of anesthesia was checked using the intertoe reflex. Thilo-Tears Gel (Alcon Deutschland GmbH, Aschaffenburg, Germany) was applied to the animals to protect the cornea. Mice were then infected intranasally with wild-type *S.pn.* serotype 2 (ST2; NCTC7466) at an infectious dose of 5 × 10^6^ colony-forming units (CFU) suspended in 20 μL 1x PBS (Thermo Fisher Scientific, Waltham, MA, USA), as previously described [17] and monitored for changes in body weight and temperature every 12 h starting from 24 h post-infection (hpi). Control (sham) mice received 20 μL volume of 1x PBS. Infected mice were sacrificed at 24 and 48 hpi due to the capacity of *S.pn.* ST2 to provoke the development of severe lung infections that could progress to sepsis within that period [17]. At the corresponding analysis time points, mice were deeply anesthetized by intraperitoneal administration of ketamine (200 mg/kg of body weight; CP-Pharma, Burgdorf, Germany) and xylazine (20 mg/kg of body weight; CP-Pharma, Burgdorf, Germany) diluted in sterile isotonic saline (B. Braun, Melsungen, Germany) and checked for loss of intertoe reflexes prior to dissection.

### 2.3. Blood Sampling

Blood was drawn into EDTA-capillary tubes (Sarstedt AG, Nümbrecht, Germany) and measured on the scil Vet Animal Blood Counter (scil animal care company GmbH, Viernheim, Germany) to record circulating leukocytes. The same samples were utilized to determine blood bacterial burden and then centrifuged to obtain plasma (2000× *g*, 10 min, 4 °C).

### 2.4. Bronchoalveolar Lavage Cells and Fluid

Perfused lungs were flushed through the trachea twice with 0.8 mL 1x PBS supplemented with protease inhibitors (Roche, Basel, Switzerland) and centrifuged to obtain bronchoalveolar lavage (BAL) cells, which were suspended in FACS buffer (1x PBS + 0.2% bovine serum albumin (BSA); Sigma-Aldrich, St. Louis, MO, USA) and stored on ice for flow cytometry. The supernatant (BAL fluid) was used for further analyses of pro-inflammatory mediators. 

### 2.5. Co-Isolation of Murine Pulmonary Epithelial Cells and Lung Neutrophils

Murine lungs were perfused through the right heart ventricle with 1x PBS followed immediately by dispase (Corning GmbH, Wiesbaden, Germany). BAL was performed as previously described, followed by intratracheal application of 0.7 mL dispase and 0.5 mL 1% UltraPure low melt agarose (Invitrogen, Carlsbad, CA, USA) into the lungs. After solidification of the agarose, lungs were removed, washed in 1x PBS and then digested with MACS buffer (1x PBS + 0.5% BSA) supplemented with collagenase type II (Biochrom GmbH, Berlin, Germany) and DNase I (AppliChem, Darmstadt, Germany) at 37 °C in an orbital shaker for 30 min. The digested lung suspension was then passed through a 70 µm filter, washed with MACS buffer and centrifuged. After treating the pellet with red blood cell lysis solution, cells were counted using a hemocytometer (Marienfeld, Lauda-Königshofen, Germany) and cell density was adjusted to 10^8^ cells/mL. Total lung cells were stained with 10 µL biotinylated monoclonal antibodies (mAb) against Ly6G (1A8, BioLegend, San Diego, CA, USA) per 10^7^ cells to label PMN, vortexed and incubated for 15 min at room temperature. Meanwhile, 10 µL MagniSort Negative Selection Beads (Invitrogen, Carlsbad, CA, USA) per 10^7^ cells were washed 3 times with MACS buffer in the presence of a magnetic field prior to resuspending the beads in 100 µL MACS buffer per 10^7^ cells. Upon completion of the incubation time, biotinylated lung cells were washed with MACS buffer, centrifuged, the supernatant discarded, and the pellet resuspended in 100 µL MACS buffer per 10^7^ cells. The resuspended cellular pellet was incubated with the beads suspension for 5 min to allow for biotin-streptavidin binding, followed by magnetic separation with a DynaMag-2 magnet (Invitrogen, Carlsbad, CA, USA). Biotinylated PMN were positively selected towards the magnet whereas all non-Ly6G expressing cells remained in the negative fraction supernatant. The negative fraction supernatant was carefully removed into a 15 mL centrifuge tube and the same procedure repeated using a biotinylated mAb cocktail against CD31 (MEC 13.3, BD, Heidelberg, Germany), CD16/32 (2.4G2, BD, Heidelberg, Germany) and CD45 (30-F11, BD, Heidelberg, Germany) to label the remaining cell types (endothelial and hematopoietic cells). Murine pulmonary epithelial cells were isolated from the PMN-depleted cell suspension by negative selection using the DynaMag-2 magnet. 

### 2.6. Bone Marrow Neutrophil Isolation

The femur and tibia were flushed with 1x PBS into a conical tube containing MACS buffer, and the total bone marrow cell suspension was centrifuged (470× *g* for 5 min, 4 °C). The pellet was cleared of red blood cells using erythrocyte lysis solution, as described earlier. The remaining bone marrow cells were counted using a hemocytometer and the cell density adjusted to 200 µL MACS buffer per 10^8^ cells. Murine bone marrow neutrophils (BM-PMN) were then isolated via positive selection using the Anti-Ly-6G MicroBead Kit (Miltenyi Biotec, Bergisch Gladbach, Germany) according to the manufacturer’s instructions. 

### 2.7. Phagocytosis and Killing Assay

In order to evaluate the phagocytosis and killing capacity of neutrophils, WT and miR-223^−/−^ BM-PMN were isolated using the Anti-Ly-6G MicroBead Kit (Miltenyi Biotec, Bergisch Gladbach, Germany) and seeded in 1.5 mL microcentrifuge tubes at a density of 4 × 10^6^ cells/mL in HBSS + Ca^2+^
*+* Mg^2+^ (HBSS++; Thermo Fisher Scientific, Waltham, MA, USA). GFP-expressing *S.pn.* ST2 were opsonized using serum derived from WT or miR-223^−/−^ mice for 30 min at 37 °C followed by infection of WT and miR-223^−/−^ BM-PMN at multiplicity of infection (MOI)10 and MOI100. Infected BM-PMN were incubated for 1h to allow for phagocytosis of opsonized GFP+ *S.pn.* ST2, followed by treatment with 100 µg/mL final concentration of gentamicin (Sigma-Aldrich, St. Louis, MO, USA) for 5 min to clear extracellular bacteria. BM-PMN were then gently washed twice with HBSS++ followed by additional incubation at 37 °C for 1 h and 2 h prior to evaluating engulfed GFP-expressing *S.pn.* ST2 in live BM-PMN using flow cytometry.

### 2.8. Single Cell Lung Suspensions

Lungs were perfused through the heart, followed by removal and manual mincing of the left lobe in a tissue culture dish containing RPMI 1640 medium (PAA Laboratories GmbH, Pasching, Austria) supplemented with 10% FCS (CAPRICORN Scientific, Ebsdorfergrund, Germany), 1% HEPES (Life Technologies, Darmstadt, Germany), 1% L-glutamine (Life Technologies, Darmstadt, Germany), DNase I (AppliChem, Darmstadt, Germany) and collagenase type II (Biochrom GmbH, Berlin, Germany). The lung suspension was incubated in a rocking water bath at 37 °C for 30 min. Digested lung suspensions were then passed through 100 μm filters and red blood cells cleared by exposure to erythrocyte lysis solution (Miltenyi Biotec, Bergisch Gladbach, Germany) for 2 min. The lung suspension was then centrifuged and resuspended in FACS buffer on ice until further analyses. 

### 2.9. Flow Cytometry

Cell surface antigen staining was performed on single cell BAL suspensions to identify live PMN, alveolar macrophages (AM) and inflammatory monocyte/macrophages (iM) (Appendix A). Approximately 10^6^ BAL cells were blocked with CD16/CD32 (2.4G2, BD, Heidelberg, Germany) to block non-specific antibody binding. After 5 min, cells were stained with mAbs against CD45 (30-F11, BD, Heidelberg, Germany), CD11c (N418, eBioscience, San Diego, CA, USA), CD11b (M1/70, eBioscience, San Diego, CA, USA), F4/80 (BM8, eBioscience, San Diego, CA, USA), Ly6G (1A8, BD, Heidelberg, Germany) and Siglec F (E50-2440, BD, Heidelberg, Germany) for 20 min at 4 °C in the dark. For determination of apoptotic cells, cells were additionally stained with Fixable Viability Dye (eBioscience, San Diego, CA, USA) and Annexin V (BD, Heidelberg, Germany) according to the manufacturer’s instructions. Cells were then washed with 1x PBS, centrifuged (470× *g* for 5 min, 4 °C) and fixed in 1% formaldehyde/PBS fixation buffer. Samples were kept at 4 °C overnight in the dark. The following day, samples were washed with 1x PBS, centrifuged (470× *g* for 5 min, 4 °C) and resuspended in 100 µL FACS buffer before analyzing total numbers and frequencies of innate leukocyte populations on the FACS Canto II (BD, Heidelberg, Germany). Total cell numbers were calculated using CountBright™ Absolute Counting Beads (Thermo Fisher Scientific, Waltham, MA, USA) according to the manufacturer’s instructions. Cells were analyzed using FlowJo (Ashland, OR, USA) software (v. 10).

### 2.10. Histology

Mice were sacrificed at 48 hpi according to previously described guidelines [18,19], immersed in 4% buffered formaldehyde solution and fixed for 24 h at room temperature. Lungs were cut into 2 μm sections and stained with hematoxylin and eosin (H&E). Lung inflammation was characterized based on parameters such as neutrophilic infiltration into the airways and lung parenchyma, pleuritis, steatitis, perivascular edema, hemorrhages and the affected lung area. Parameters were scored 1–4 (1: minimal, 2: light grade, 3: middle grade, and 4: high grade).

### 2.11. RNA Isolation, Reverse Transcription and Quantitative PCR of Murine Lungs and Primary Cells

Total RNA from murine lung homogenates and primary cells stored in TRIzol reagent were extracted according to the manufacturer’s instructions. The concentration of extracted RNA was measured using the Nanodrop 2000 (Thermo Fisher Scientific, Waltham, MA, USA), followed by dilution of the RNA in diethylpyrocarbonate (DEPC)-treated water (Invitrogen, Carlsbad, CA, USA) at a concentration of 10 ng/µL. An amount of 50 ng of RNA was reverse transcribed to cDNA using the TaqMan™ MicroRNA Reverse Transcription Kit (Applied Biosystems, Waltham, MA, USA) according to the manufacturer’s instructions, utilizing pre-designed stem-loop primers for miR-223-3p (Assay ID 002295, TaqMan™ MicroRNA Assay, Applied Biosystems, Waltham, MA, USA) and snoRNA202 (Assay ID 001232, TaqMan™ microRNA Control Assay, Applied Biosystems, Waltham, MA, USA). Samples were mixed and placed in a thermocycler (Biometra, Göttingen, Germany), followed by incubation for 30 min at 16 °C, 30 min at 42 °C and 5 min at 85 °C. An amount of 5 ng of cDNA template from target and endogenous control genes was amplified for each sample using the TaqMan™ Fast Advanced Master Mix (Applied Biosystems, Waltham, MA, USA) protocol according to the manufacturer’s instructions and, analysis was performed on the StepOnePlus™ Real-Time PCR System (Applied Biosystems, Waltham, MA, USA). Relative quantification (RQ) of miR-223 expression was calculated using the 2^-ddCT^ method [20] utilizing naïve or sham-infected cells/tissues as calibrator and snoRNA202 as the housekeeping gene. 

### 2.12. RNA Isolation, Reverse Transcription and Quantitative PCR of miR-223 in Serum and Plasma of Humans and Mice

Serum miR-223-3p (Assay ID 477983_mir, TaqMan™ Advanced miRNA Assay (Applied Biosystems, Waltham, MA, USA)) was analyzed in healthy controls and pneumococcal pneumonia patients following RNA isolation (mirVana™ PARIS™ Kit (Life Technologies, Darmstadt, Germany)), reverse transcription (TaqMan™ Advanced miRNA cDNA Synthesis Kit (Applied Biosystems, Waltham, MA, USA)) and qRT-PCR. RNA was extracted from 200 µL volume of patient serum or mouse plasma according to the manufacturer’s instructions following the addition of 75 pg exogenous 5′ phosphorylated cel-miR-39-3p (Assay ID 478293_mir, TaqMan™ Advanced miRNA Assay (Applied Biosystems, Waltham, MA, USA)) into the biosamples. An amount of 2 µL of extracted RNA was used for poly(A) tailing reaction, followed by adaptor ligation and reverse transcription, as per the manufacturer’s instructions. cDNA template from target, endogenous control and exogenous control genes were amplified for each sample using the TaqMan™ Fast Advanced Master Mix (Applied Biosystems, Waltham, MA, USA) protocol according to the manufacturer’s instructions, and analysis was performed on the QuantStudio™ 5 System (Applied Biosystems, Waltham, MA, USA). Relative abundance of miR-223 in the serum of CAP patients was calculated using the 2^−ddCT^ method [20]; utilizing sera of healthy subjects as calibrators and miR-24-3p (Assay ID 477992_mir, TaqMan™ Advanced miRNA Assay (Applied Biosystems, Waltham, MA, USA)) as the housekeeping gene.

### 2.13. Single Cell Transcriptomics

Viable cells were isolated from whole lungs of mice following intranasal infection with *S.pn.* ST2 or sham at 24 hpi. Single-cell suspensions were then pooled by experimental groups and dead cell removal was performed by magnetic cell sorting (Miltenyi Biotec, Bergisch Gladbach, Germany). Barcoding and library construction was performed according to the manufacturer’s instructions (10X Genomics, Pleasanton, CA, USA), followed by quality control checks via Fragment Analyser (Agilent, Waldbronn, Germany) using the NGS Fragment Kit (1-6000bp) and Qubit fluorometric quantification with the dsDNA HS assay kit (Invitrogen, Carlsbad, CA, USA). Next-generation sequencing was performed according to the Illumina protocol on a NextSeq500 device (Illumina, San Diego, CA, USA) with a High Output v2 Kit (150 cycles) and the recommended sequencing conditions (read1: 26nt, read2: 98nt, index1: 8nt, index2: n.a.). Output data was aligned to the mouse genome build mm10/GRCm38 followed by applying Cell Ranger functions mkfastq and Cell Ranger count (10X Genomics Cell Ranger Software, Pleasanton, CA, USA). Downstream analyses were performed using software R (Vienna, Austria) version 4.1.1 with packages by Bioconductor [21] and Seurat v. 4.1.1 [22]. Low-quality cells were removed, keeping only cells with less than 30% mitochondrial genes, at least 300 different genes, and RNA counts between 300 and 35,000. Individual expression matrices of experimental groups were normalized, variance stabilized and scaled by the SCT transform method [23]. Four thousand variable features were used in the integration of all datasets. For this, normalized expression data of the cells was reduced to 50 PCA components and projected into a shared embedding using the R package Harmony v 1.0 [24]. After applying uniform manifold approximation and projection (UMAP), cells were clustered by applying the Louvain algorithm. Multiplets were excluded after predicting them with DoubletFinder v 2.0.3 [25]. This strategy minimizes batch effects and technical noise, while favoring grouping of cells by cell type. Cell populations were assigned through RNA marker gene allocation from databases [26,27] and literature (annotated in Appendix A); namely, alveolar macrophages (*Marco* [28]); alveolar & interstitial monocyte-derived macrophages (*Adgre1* [29]); dendritic cells (*Flt3* [30,31]); neutrophils (*S100a8* [32]); eosinophils & alveolar macrophages (*Siglecf* [33]); AT1 (*Akap5* [31]); AT2 (*Lamp3* [31] and *Sftpc* [34]; ciliated cells (*Foxj1* [31]); endothelial cells (*Cdh5* [35]); lymphatic endothelial cells (*Mmrn1* [36]); fibroblasts (*Inmt* [36]); smooth muscle cells (*Acta2* [36]); pericytes (*Cox4i2* [37]); mesothelial cells (*Msln* [36]); NK cells (*Ncr1* [38]); T cells (*Cd3e* [39]); and B cells (*Cd79a* [40]). Pathway enrichment analysis of significantly differentially expressed granulocyte genes (minimum fold change 1.3) was done with the R-package gprofiler2 v 0.2.1 [41] (Appendix A).

### 2.14. Statistical Analyses

Following determination of Gaussian distribution via the D’Agostino-Pearson omnibus normality test, an unpaired *t*-test or Mann-Whitney U test was performed to analyze the statistical significance between two independent groups of mice. A 2-way ANOVA with Šidák’s multiple comparisons test was performed when comparing independent groups across multiple time-points in the murine model of pneumonia. Comparison of survival curves was performed using the Log-rank (Mantel-Cox) and Gehan-Breslow-Wilcoxon tests. Differential expression analysis was performed using the Wilcoxon signed-rank test and adjusted P-values were computed via the Bonferroni multiple testing correction for the identified cell types and genes following scRNA-seq of murine lungs. Correction for multiple testing in pathway enrichment analysis was done as previously described [42] and considered an experiment-wide threshold of alpha = 0.05. The proportional odds model was utilized to compute the correlation between serum miR-223 relative abundance and disease severity in CAP patients. Statistical analyses were performed using either GraphPad Prism (San Diego, CA, USA) software version 9.2.0 or software R (Vienna, Austria) version 4.1.1. 

## 3. Results

### 3.1. miR-223 Is Reduced in Human Serum during Bacterial Pneumonia and Transiently Increases in Murine Lungs upon Pneumococcal Infection

We first evaluated circulating cell-free miR-223 in the sera of 92 CAP patients as well as in 50 healthy blood donors. We detected a ~31.5% reduction in levels of miR-223 by qRT-PCR analyses in sera of CAP patients as compared to those of healthy control subjects (Figure 1A). Moreover, increased disease severity in CAP patients, evaluated using the CURB-65 score, in addition to enhanced blood C-reactive protein (CRP) concentrations recorded in CAP patients, were associated with reduced miR-223 levels in the sera of pneumonia patients (β = −0.91, r = −0.2439 respectively; Figure 1A). Taking into account the median value of 0.6455 for serum miR-223 relative abundance in CAP patients, CRP concentrations were mildly elevated in CAP patients in which serum miR-223 relative abundance was recorded below 0.6455 (*p* = 0.0118, Table 1); a finding which was not attributable to differences in circulating neutrophil numbers (Appendix A, Table 1). In mice, no changes in circulating miR-223 expression were observed in the plasma (Appendix A) or in isolated bone marrow neutrophils (BM-PMN) following *S.pn.* stimulation (Figure 1B). However, miR-223 was upregulated in the lungs and pulmonary PMN 24 hpi compared to sham (1x PBS)-infected, or naïve mice (Figure 1B). No significant upregulation of miR-223 in alveolar epithelial cells (AEC) 24 or 48 hpi (Appendix A) was observed.

### 3.2. Absence of miR-223 Renders Mice Prone to Exacerbated Lung Inflammation

At the onset of pneumonia, WT and miR-223^−/−^ mice responded similarly to *S.pn.* infection with respect to body temperature and body weight 24 hpi. However, with inflammation peaking at 48 hpi, miR-223^−/−^ mice displayed greater weight loss (Figure 2A) despite similar bacterial burden recorded in the lungs, BAL, spleen and blood (Figure 2B). Histopathological analyses of infected murine tissues revealed a generally increased inflammatory state of the lungs 48 hpi in the absence of miR-223, characterized by excessive neutrophilic infiltrates, exacerbated pleuritis, steatitis, perivascular edema and hemorrhages (Figure 2C). However, this did not significantly affect overall survival (Figure 2D).

### 3.3. miR-223 Regulates Pulmonary Neutrophil Migration, Persistence and Apoptosis

Twenty-four hours following intranasal *S.pn.* ST2-infection, WT and miR-223^−/−^ mice exhibited similar numbers and frequencies of PMN in blood, BAL and lungs (Appendix A). At 48 hpi however, as WT PMN frequencies declined, miR-223^−/−^ PMN remained elevated in the BAL and lungs (Figure 3A), concomitant with the deterioration of physiological parameters (Figure 2A). Likewise, miR-223^−/−^ mice displayed significantly lower proportions of neutrophils bearing early apoptosis markers in the BAL compared to WT mice (Figure 3B). Moreover, 48 hpi PMN recordings in blood of miR-223-deficient mice were significantly higher than those of WT mice (Appendix A). PMN-attracting chemokine concentrations were found to match the neutrophilia exhibited in histopathological and flow cytometric analyses (gating strategy depicted in Appendix A), as seen by enhanced CXCL1 and CXCL5 in the BAL of miR-223^−/−^ mice 48 hpi (Figure 3C). We recorded no differences in Ly6C^hi^ monocyte-derived inflammatory macrophages (Ly6C^hi^ iM) while recording a modest decline in miR-223^−/−^ alveolar macrophage (AM) frequencies in the BAL 48 hpi (Appendix A), most likely due to the enhanced neutrophilic influx. Furthermore, flow cytometry revealed that neither BAL nor lung neutrophils displayed differential expression of CD11b, as indicated by the mean fluorescence intensity (MFI) of CD11b in WT and miR-223^−/−^ PMN 48 hpi (Appendix A). We also did not observe any differences in phagocytosis or killing of GFP-expressing *S.pn.* ST2 by bone marrow neutrophils (Appendix A).

### 3.4. Absence of miR-223 Triggers Elevated Production of Inflammatory Mediators

Considering the prolonged neutrophil response exhibited by miR-223^−/−^ mice, we next investigated the degree of lung inflammation in WT and miR-223^−/−^ mice by quantification of pro-inflammatory cytokine concentrations during the course of pneumonia. We detected similar concentrations of pro-inflammatory cytokines such as IL-6, TNF-α, IFN-γ, IL-1α and neutrophil myeloperoxidase (MPO) in the BAL of miR-223^−/−^ mice compared to WT mice at 24 hpi (Appendix A). At 48 hpi, however, levels of the same pro-inflammatory cytokines and neutrophil MPO were elevated in the BAL of miR-223-deficient animals (Figure 4). 

### 3.5. miR-223 Regulates Genes Involved in Inflammation, Granulocyte Maturation and Antibacterial Defense

From the various cellular populations clustered following scRNA-seq analyses (Figure 5A, Appendix A), granulocytes of miR-223^−/−^ mice contained the highest number of differentially expressed genes that underwent a fold-change of 1.3 or more following *S.pn.* infection, in comparison to the corresponding cells of *S.pn.*-infected WT mice (Figure 5B). Genes identified to be differentially expressed in miR-223^−/−^ granulocytes include predicted (*Atp1b1*, *Ctsd*, *Cybb*, *Hmgb2*, *Lst1*, *Parp9*, *Ywhah*) and validated (*Cxcl2*) targets of miR-223, as identified via miRTarBase [43], miRDB [44], TargetScan v8. [45] and miRPathDB 2.0 [46]; in addition to Spi1, an inducer of miR-223 [47] (Figure 5C). Additionally, when analyzing differential gene expression only amongst granulocytes of *S.pn.*-infected WT and miR-223^−/−^ mice, we identified increased expression of the validated miR-223 target *Stat3*, in addition to *Mmp8* and *Nfkbiz* (Appendix A). Collectively, the differentially expressed genes were implicated in cellular processes such as inflammation (*Il1b*, *Nfkbia*, *il6ra*), chemotaxis (*Cxcl2*, *Cxcl3*), granularity and enzymatic digestion (*Ctsb*, *Ctsd*, *Lyz2*, *Lcn2*) and granulocyte maturation (*Mxd1*, *Prdx5*), as depicted in the representative dot plot (Figure 5C; *p*-values provided in Appendix A). Pathway enrichment analysis indicated antimicrobial peptides and the humoral immune response as the top two biological processes that were enriched in miR-223^−/−^ granulocytes following *S.pn.* infection (Figure 5D, Appendix A). These findings collectively point towards a hyper-mature phenotype of miR-223^−/−^ PMN.

## 4. Discussion

In this study, we devised the anti-inflammatory role of miR-223 during pneumococcal pneumonia in humans and mice. Circulating miR-223 was reduced in the sera of CAP patients compared to those of healthy subjects. Increased disease severity, in addition to elevated blood CRP concentrations recorded in CAP patients, correlated to lower relative abundance of serum miR-223 in the sera of CAP patients; independent of circulating neutrophil numbers. Furthermore, we found miR-223 to be induced in a time- and cell-dependent manner in our murine model of pneumococcal pneumonia, and in that way regulating granulocyte maturation, chemotaxis and production of pro-inflammatory mediators during pneumococcal pneumonia. Following *S. pneumoniae* infection, lung-infiltrating PMN in WT mice were found to endogenously and transiently upregulate miR-223, while miR-223 deficient mice exhibited exacerbated lung pathology with sustained neutrophil presence in alveolar spaces and lung tissue. Even though miR-223 has been studied extensively in the context of chronic [16] and sterile inflammation [48,49,50], its relevance in acute bacterial lung infections has not yet been sufficiently elucidated, especially considering the significant involvement of PMN in antibacterial responses as well as in host-mediated tissue injury. Circulating CRP and IL-6 are established biomarkers for bacterial infections but have limited value for characterizing disease severity or for making specific prognoses for individual cases. However, recent studies have illustrated the potential of signature miRNAs in biofluids and organ specimens as indirect markers for predicting the duration of the hospital stay of pneumonia patients. Other miRNAs have been implicated to act as indicators of CAP and sepsis [51,52]; with the limitation that sample sizes in the respective studies were small and the latter lacked complementary mechanistic investigations. Results conflicting to ours have been previously reported in an observational study, whereby miR-223 expression was described to be enhanced in the plasma of pneumonia and sepsis secondary to pneumonia cohorts [53]. This discrepancy, however, could be explained from several standpoints. Firstly, the aforementioned study describes miR-223 expression in plasma rather than serum, which may result in higher miRNA concentrations compared to that of serum [54]. Secondly, the authors normalized changes in gene expression to a different housekeeping gene (miR-16). Most importantly, the study does not indicate the types of pathogens detected in their CAP cohort, whilst the presence of *S. pneumoniae* was confirmed in all subjects of our cohort (Table 2). Consequently, it could be inferred that there is a need for more standardized methodology and in-depth investigations of mediators of RNA interference and gene-silencing within the scope of biomarker discovery, pathogen-driven inflammation and organ injury.

Of note, we did not record any changes in miR-223 expression in the plasma of sham- or *S.pn.*-infected mice. Considering the increased leukocyte chemotaxis to inflamed tissue during bacterial infection, similar levels of miR-223 expression in plasma of sham- and *S.pn.*-infected mice point towards a lack of miR-223 release into the circulation in mice during pneumococcal pneumonia. However, the significantly higher sample size in our CAP cohort, coupled to the analysis of serum rather than plasma, render any direct comparison of circulating miR-223 expression in humans and our murine pneumonia model inconclusive, which is a limitation of our study. Furthermore, we did not detect changes in miR-223 expression in isolated BM-PMN following pneumococcal stimulation. We did, however, observe increased miR-223 expression in whole lungs and in sorted pulmonary PMN of *S.pn.*-infected WT mice 24 hpi, but we were unable to detect significant upregulation of miR-223 in sorted AEC. This contradicted previous findings which reported that miR-223 transfer from PMN to AEC reduces pulmonary inflammation induced by mechanical ventilation or intratracheal *S. aureus* administration [55]. In the same study, overexpression of miR-223 in mice was described to reduce *Il6*, *Cxcl1* and *Tnfa* transcripts; an effect assumed to be mediated by miR-223-targeted inhibition of *Parp1* and subsequently reducing acute lung injury following mechanical ventilation or *S. aureus* infection [55]. Pulmonary pathology in our murine model of pneumococcal pneumonia, however, is likely driven independent of *Parp1* as indicated by the absence of differential gene regulation of its gene product in our murine model of intranasal infection with *S.pn.* ST2. Instead, we found the *Parp9* transcript to be expressed only moderately yet significantly higher in the lungs of miR-223^−/−^ mice infected with *S.pn.* as compared to expression in WT mice. PARP-9 was recently described to be a non-canonical sensor for RNA viruses in dendritic cells, aiding the production and amplification of type 1 interferons [56]. The induction of *PARP-9* has been reported to be influenced by IFN-γ [57], the concentration of which was elevated in the BAL of *S.pn.*-infected miR-223^−/−^ mice 48 hpi.

Furthermore, a recent study characterizing neutrophil sub-populations across COVID-19 ARDS and bacterial ARDS patients detected IFN active (*IFITM1*, *IFITM2*, *IFI6* expression), prostaglandin (PG) active (*PTGER4*, *PTGS2* expression) and bacterial-enriched (*CD83*, *CD177*, *PLAC8* expression) transcriptional states in blood samples of these patients [58]. *Cd177* and *Plac8* transcripts were significantly upregulated while *Ifitm2* was downregulated compared to WT mice in our scRNA-seq analyses of lung neutrophils of *S.pn.*-infected miR-223^−/−^ mice. This finding shows a similarity of miR-223^−/−^ pulmonary PMN with the bacterial-enriched phenotype seen in bacterial ARDS patients. We did not detect significant differences in bacterial burden between WT and miR-223^−/−^ mice despite *Plac8* being acknowledged to have antibacterial properties, as revealed by the increased bacterial burden in its absence following *K. pneumoniae*-induced acute peritonitis [59]. Nevertheless, we demonstrated that the comparable levels of bacterial burden between WT and miR-223^−/−^ mice were not due to a defect in phagocytosis or killing of bacteria by miR-223^−/−^ neutrophils, despite the enhanced neutrophil infiltration recorded in miR-223^−/−^ mice. Although a pioneering study characterizing miR-223 previously described enhanced fungicidal capacity of miR-223^−/−^ PMN [15], this was not evident in relation to pulmonary bacterial pathogens; as evident in a similar study of acute lung injury reporting analogous findings of comparable BAL bacterial burden of *S. aureus*-infected WT and miR-223^−/−^ mice, despite an enhanced neutrophil response in the latter experimental group [55]. Although we were able to identify that *S.pn.*-infected miR-223^−/−^ lung neutrophils downregulated genes involved in antioxidation while upregulating a number of genes with antimicrobial properties, the reason for the lack of reduction in bacterial load in miR-223^−/−^ mice remains speculative. *Mxd1* is known to be a transcriptional regulator of PMN cell-fate transitions and is expressed in the latter stages of neutrophil maturation, but not in neutrophil progenitors [60]. Furthermore, CD52 is a glycosylphosphatidylinositol-anchored protein expressed on myeloid cells and has been shown to act as an anti-inflammatory mediator leading to innate immune cells dampening TLR and TNF signaling [61], while also being predicted to be regulated by *Mxd1* according to gene regulatory network (GRN) analyses [60]. PRDX5 has been shown to play a protective role as an antioxidant in different tissues in inflammatory processes [62] and its gene transcript is highly expressed during the transition of hematopoietic stem cells to mature neutrophils [60], while its expression was shown to dramatically decrease in mature blood neutrophils [63]. Thus, our findings of upregulated *Mxd1* and downregulated *Prdx5* in lung neutrophils of *S.pn.*-infected miR-223^−/−^ mice (compared to that of *S.pn.*-infected WT mice) suggest a transcriptionally hyper-mature state in lung neutrophils lacking miR-223. A recent study described genes involved in granularity (*Lcn2*, *Chil3*) and antimicrobial effects (*Lyz2*, *Hmgb2*) to be highly expressed in developing neutrophils in the bone marrow, with their expression levels reduced upon maturation and release of neutrophils into the blood [63]. Surprisingly, only *Hmgb2* was downregulated in miR-223^−/−^ lung neutrophils while *Lcn2*, *Chil3* and *Lyz2* were expressed significantly higher following *S.pn.* infection as compared to WT neutrophils. Furthermore, genes such as *Il1b* and *Fth1* were shown to be highly expressed only in mature blood neutrophils [63], which matches the abundant expression levels recorded in miR-223^−/−^ lung neutrophils following *S.pn.* infection. *Il1b* has also been shown to induce increased expression of *Nfkb* inhibitors such as *Nfkbia* and *Nfkbiz* in primary murine hepatocytes in a model of LPS-mediated inflammation [64], which could explain their increased expression in lung neutrophils of *S.pn.* ST2-infected miR-223^−/−^ mice, despite the absence of differential regulation of *Nlrp3*. Furthermore, *Nfkbiz* was demonstrated to induce the transcription of antibacterial *Lcn2* through interaction with NF-κB p50 in the *Lcn2* promoter region, dependent on the pyrimidine-rich responsive site (CCCCTC) [65]. MMP8, a collagenase produced by neutrophils that functions to degrade extracellular matrix [66], was shown to be a crucial mediator of ventilator-induced lung injury (VILI), as corroborated by reduced neutrophilic infiltration, lower IFN-γ and CXCL2 concentrations, improved gas exchange and decreased lung edema and histopathology in mechanically ventilated *Mmp8*^−/−^ mice [67]. Cell signaling via STAT3, a validated target of miR-223, has been described as being induced by IL-6 in the lungs [68]. In these experiments, absence of *Stat3* resulted in lower numbers of alveolar neutrophils and a higher bacterial load following intratracheal *E. coli* administration. Moreover, the degree of lung injury was increased at 48 hpi [68]. Taken together, our findings strongly support a model in which pneumococcal pneumonia triggers a hyper-mature phenotype of miR-223^−/−^ neutrophils, encompassing a transcriptional state exhorting continued expression of granularity and pro-inflammatory genes which is not evident in WT lung neutrophils.

Although the survival of mice with pneumonia was not impacted by miR-223 deficiency, absence of miR-223 rendered mice susceptible to deteriorating physiological conditions characterized by prolonged neutrophil extravasation, exacerbated histopathology and pro-inflammatory cytokine production. This effect was independent of alveolar or monocyte-derived inflammatory macrophage involvement, as demonstrated by their largely matching cell numbers and frequencies in WT and miR-223^−/−^ mice coupled with the lack of differential gene expression in the absence of miR-223 in the aforementioned cell types, as revealed in our scRNA-seq dataset. This argument is further augmented by the fact that miR-223 expression is significantly downregulated during the transition of granulocyte/monocyte precursors to mature monocytes [15]. The persistent PMN response in miR-223^−/−^ mice correlated with pronounced CXCL5 and CXCL1 concentrations in the BAL 48 hpi, while CXCL2 concentrations remained normal in the BAL of WT and miR-223^−/−^ mice despite increased expression of *Cxcl2* mRNA transcripts in granulocytes of *S.pn.*-infected miR-223^−/−^ lung tissue 24 hpi. The highly heterogeneous nature of lung tissue [69] could help explain this discrepancy as compared to the composition of BAL cells and fluid. Although pulmonary CXCL5 is described to be exclusively produced by AEC [70], lack of enhanced miR-223 uptake by WT murine AEC post-*S.pn.* infection suggests pulmonary epithelia involvement independent of miR-223. However, different experimental models of infection and organ injury have a direct influence on the readouts and implications of miR-223-mediated inflammation: Whereas miR-223^−/−^ mice cleared *S. aureus* more efficiently from the wound site [48] in a murine model of wound healing, an effect described to be related to enhanced reactive oxygen species (ROS) production, *Mtb* infection resulted in increased bacterial burden in miR-223^−/−^ mice in a chronic disease setting [16]. Strikingly, miR-223 was described to play converse roles in different septic kidney injury models from the same study. In the absence of miR-223, kidney injury was attenuated following cecal ligation and puncture (CLP), whereas after LPS injection kidney injury was exacerbated [71]. Collectively, these highlight the importance of taking into account the experimental models of related study designs and scopes before drawing firm conclusions. 

There are certain limitations to our study: So far, we were not able to specify the exact source of the induction and downregulation of miR-223 in our analyses. We did, however, detect a modestly higher expression of *Spi1* in WT PMN, which is a transcription factor and one of the master regulators of miR-223 [47]. Additionally, we performed scRNA-seq with a relatively small sample sizes, hence subsequent strict statistical corrections for multiple testing were performed for differential gene expression analyses. These, coupled with the low RNA and high RNase content of PMN [63], however, indicate that involvement of other miR-223 regulators such as C/EBPα, C/EBPβ and NFI-A [72] cannot be ruled out.

We have previously demonstrated the significance of early antibiotic intervention in preventing systemic inflammation. Administering of antibiotics drastically improves survival of mice infected with pneumococci [73]; however, reinforcement of individualized therapies—including the discovery of novel prognostic and diagnostic molecular candidates—remains a forefront priority in clinical research. As miRNAs have the capacity to modulate targets conventionally dubbed “undruggable” [74], their potential for future clinical use as a biomarker for neutrophilic inflammation and disease severity in CAP could help reduce the burden of pneumonia in the future. 

## 5. Conclusions

To conclude, we demonstrate that microRNA-223 plays a role in human pneumonia whereby reduced levels in patient serum are associated with increased disease severity. Additionally, we describe exacerbated pulmonary inflammation and a dysregulated transcriptome in infected microRNA-223 knockout mice and murine lung neutrophils, respectively. Our findings emphasize two major points regarding the cellular immune response to pneumococcal pneumonia: First, there is still a great need for a more thorough understanding of neutrophils and their behavior during inflammation. Second, more research needs to be performed on both existing and novel experimental interventional approaches for CAP in the clinic. Our results highlight the significance of RNA interference in regulating pulmonary inflammation during pneumococcal pneumonia. They are a step on the way to further understand and elucidate the molecular mechanisms involved in CAP, and thus may help advance possible novel treatment options for severe lung disease in the future.

## Figures and Tables

**Figure 1 cells-12-00959-f001:**
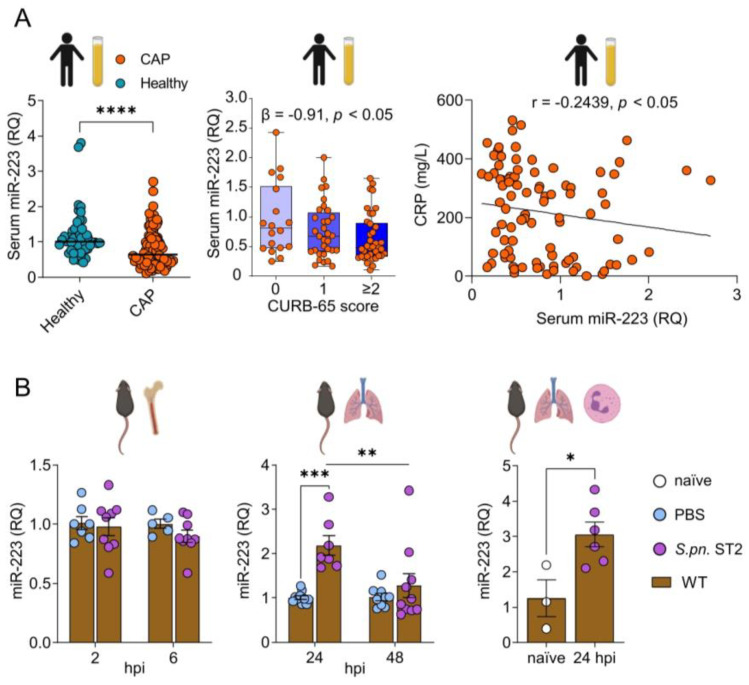
miR-223 is differentially regulated in human serum, murine lung tissue and pulmonary neutrophils during pneumococcal pneumonia. (**A**) miR-223 was quantified in the serum of healthy subjects and CAP patients using qRT-PCR (*n* = 50 and *n* = 92, respectively). CAP patients were grouped by CURB-65 scores and the proportional odds model was utilized to analyze the correlation of serum miR-223 relative abundance and disease severity. Spearman correlation was performed to determine the relationship between CRP and serum miR-223 in CAP patients. (**B**) Expression of miR-223 in vitro and in vivo in WT naïve, sham- and *S.pn.* ST2-infected mice. Isolated BM-PMN were stimulated 2 and 6 hpi with *S.pn.* ST2 (*n* = 9) or sham (*n* = 5 and *n* = 7). WT mice were intranasally infected with *S.pn.* ST2 (*n* = 7; 24 hpi, *n* = 10; 48 hpi) or sham (*n* = 11; 24 hpi, *n* = 9; 48 hpi), followed by quantification of miR-223 in whole lungs. Expression of miR-223 was also determined in sorted lung PMN (*n* = 3; naïve, *n* = 6; 24 hpi *S.pn.* ST2). (**A**,**B**) Mann-Whitney U test, * *p* < 0.05, **** *p* < 0.0001. (**A**) Proportional odds model, *p* < 0.05, β = −0.91; Spearman correlation, *p* < 0.05, r = −0.2439. (**B**) 2-way ANOVA/Šidák’s multiple comparisons test, ** *p* < 0.01, *** *p* < 0.001. Data in (**A**,**B**) display individual values and median or mean. Error bars indicate (A) minimum to maximum values (CURB-65 plot) and SEM. CAP: community-acquired pneumonia, RQ: relative quantification, CRP: C-reactive protein, CURB-65: (confusion, urea nitrogen, respiratory rate, blood pressure, 65 years of age and older), *S.pn.* ST2: *Streptococcus pneumoniae* serotype 2, hpi: hours post-infection, PBS: phosphate-buffered saline.

**Figure 2 cells-12-00959-f002:**
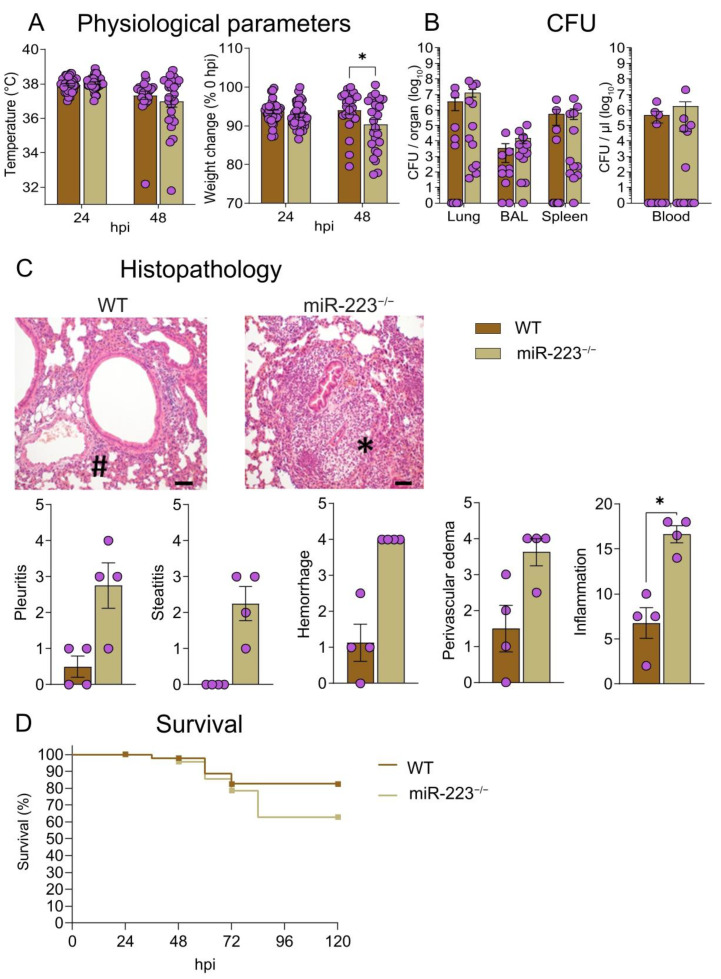
miR-223^−/−^ mice exhibit aggravated lung inflammation 48 hpi *S.pn.* ST2. (**A**) Physiological parameters, including changes in body temperature and body weight post-*S.pn*. infection in WT and miR-223^−/−^ mice. (**B**) Bacterial burden recorded in lungs, BAL, spleen and blood in WT and miR-223^−/−^ mice 48 hpi (*n* = 10–12). (**C**) Histopathological analyses of murine lungs 48 hpi. H&E staining indicates predominant neutrophilic and lymphocytic cellular infiltrates in lungs of miR-223^−/−^ and WT mice, respectively. Pictures are representative of overall inflammation in WT and miR-223^−/−^ mice. * denotes predominantly neutrophilic infiltrates coupled to pronounced edema, whilst # denotes perivascular lymphocytic cuff formation coupled to marginal edema. Pleuritis, steatitis, perivascular edema and hemorrhage scores collectively represent the overall inflammation score exhibited in the lungs of WT and miR-223^−/−^ mice 48 hpi (*n* = 4). Scale bar indicates 100 μm. (**D**) Survival curves of *S.pn.* ST2-infected WT and miR-223^−/−^ mice over the duration of 120 hpi. (**A**) 2-way ANOVA/Šidák’s multiple comparisons test, * *p* < 0.05. (**C**) Mann-Whitney U test was performed to analyze statistical significance in the overall inflammation score; * *p* < 0.05. Data in (**A**–**C**) display individual and mean values, while data in (**D**) displays censored subjects only. hpi: hours post-infection, CFU: colony-forming unit, BAL: bronchoalveolar lavage.

**Figure 3 cells-12-00959-f003:**
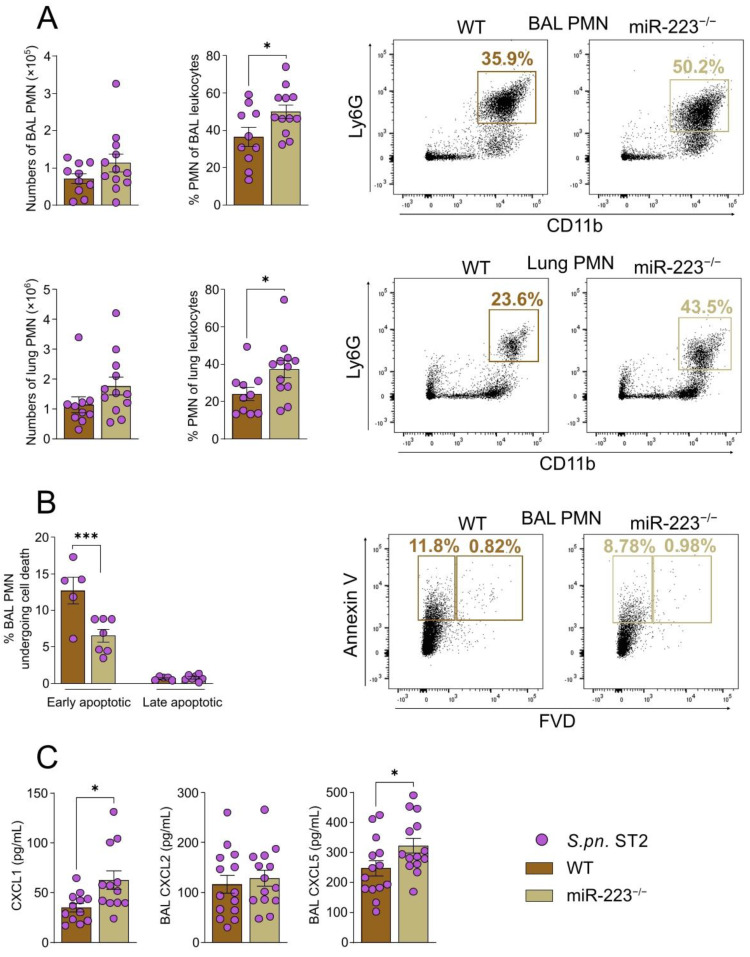
Absence of miR-223 leads to an enhanced and prolonged PMN response in the lungs and BAL of miR-223^−/−^ mice post-*S.pn.* ST2 infection. (**A**) PMN numbers, frequencies and representative dot plots in BAL and lungs 48 hpi (*n* = 10–12). (**B**) Frequencies and representative dot plots of BAL and lung PMN undergoing early and late apoptosis 48 hpi (*n* = 5–7). (C) CXCL1, CXCL2, CXCL5 chemokines quantified in the BAL of mice through ELISA 48 hpi (*n* = 10–14). (**A**,**C**) Unpaired *t*-test; * *p* < 0.05. (**B**) 2-way ANOVA/Šidák’s multiple comparisons test; *** *p* < 0.001. Data display individual values and means, error bars represent SEM. PMN: polymorphonuclear neutrophil, BAL: bronchoalveolar lavage, *S.pn.* ST2: *Streptococcus pneumoniae* serotype 2.

**Figure 4 cells-12-00959-f004:**
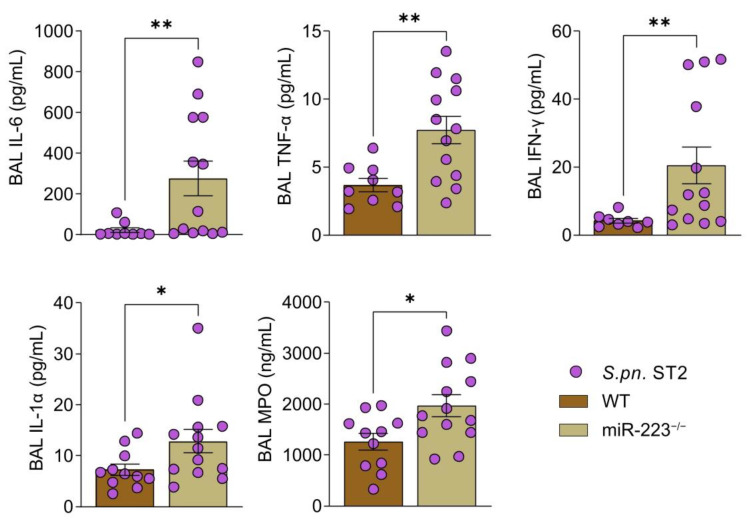
miR-223^−/−^ mice exhibit increased capacity of pro-inflammatory cytokine and chemokine production following *S.pn.* ST2 infection. Pro-inflammatory cytokines were quantified in the BAL of WT (*n* = 11) and miR-223^−/−^ (*n* = 13) mice 48 hpi using the LEGENDPlex Mouse Inflammation Panel (BioLegend, San Diego, CA, USA), whilst MPO was quantified using the MPO Mouse ELISA kit (Hycult Biotech, Uden, Netherlands). Mann-Whitney U test was performed to analyze statistical significance. * *p* < 0.05, ** *p* < 0.01; data display individual values and means, error bars represent SEM. BAL: bronchoalveolar lavage, *S.pn.* ST2: *Streptococcus pneumoniae* serotype 2, MPO: myeloperoxidase.

**Figure 5 cells-12-00959-f005:**
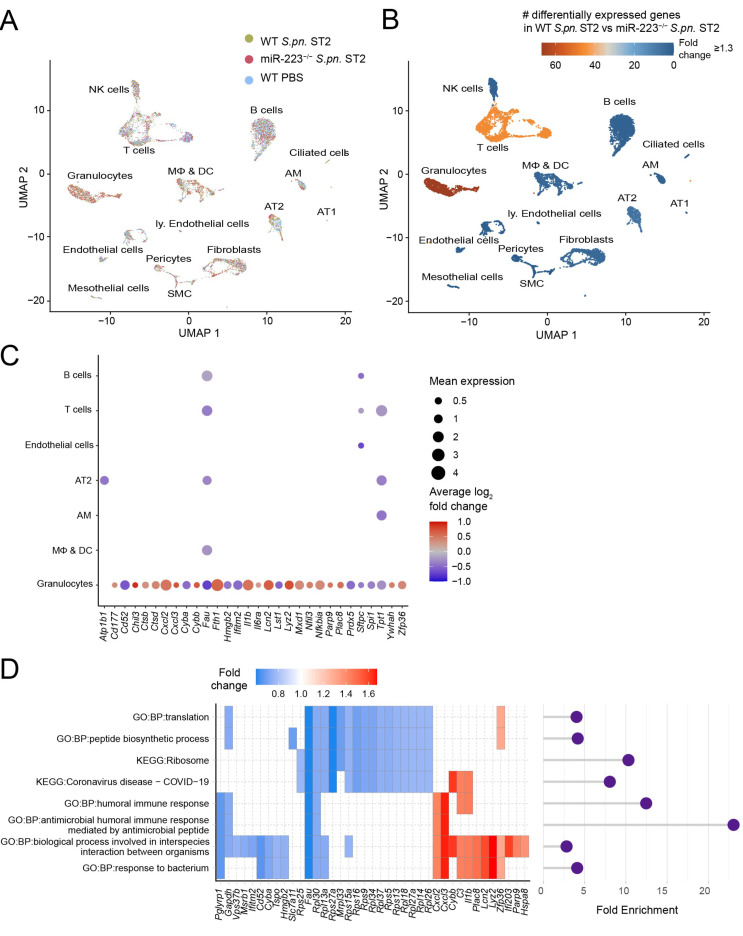
scRNA-seq of murine lungs 24h following *S.pn.* ST2 infection. (**A**) UMAP plot of identified cell populations in the lungs of WT and miR-223^−/−^ mice following *S.pn.* ST2 (WT, miR-223^−/−^; *n* = 3) or sham (PBS Ctrl, WT; *n* = 2) infection. (**B**) Number of significant differentially expressed genes (minimum fold change 1.3) in miR-223^−/−^ mice (relative to WT mice) infected with *S.pn.* (**C**) Dot plot of significantly differentially expressed genes involved in inflammation, granulocyte maturation, enzymatic digestion and antibacterial defense in WT versus miR-223^−/−^ mice. (**D**) Pathway enrichment analysis of miR-223^−/−^ versus WT lung cells sequenced following *S.pn.* infection. Coloration and point sizes indicate log_2_-transformed fold changes and mean expression levels, respectively. AT1: alveolar epithelial cells type I, AT2: alveolar epithelial cells type II, AM: alveolar macrophages, SMC: smooth muscle cells, MΦ & DC: interstitial and inflammatory macrophages/monocytes and dendritic cells, ly. Endothelial cells: lymphatic endothelial cells, NK cells: natural killer cells, GO:BP: Gene Ontology:Biological Process, KEGG: Kyoto Encyclopedia of Genes and Genomes, UMAP: uniform manifold approximation and projection.

**Table 1 cells-12-00959-t001:** Relative miR-223 abundance and clinical characteristics of the CAPNETZ cohort.

Parameter or Characteristic	miR-223 > Median Serum Abundance	miR-223 < Median Serum Abundance	*p*-Value
miR-233 relative abundance in serum	1.103 (0.8450–1.501)	0.4155 (0.3043–0.4808)	<0.0001
CRP, mg/L	183.5 (39.45–304.8)	297 (112.6–366.8)	0.0118
Blood leukocytes, cells/nL	13.60 (9.9–18.60)	16.60 (11.18–21.83)	0.3416
Blood segmented neutrophils, cells/nL	12.95 (7.308–19.04)	13.38 (9.243–18.63)	0.9950
Age	59.5 (42.5–70)	65 (44.5–75.5)	0.1563
Sex (male/female), *n*	24/22, *n* = 46	31/15, *n* = 46	0.0810

Circulating cell-free miR-223 was quantified in the serum of CAP patients, relative to that of healthy subjects. Median relative abundance of miR-223 in the serum (0.6455) was determined following normalization to circulating miR-24. Data displayed in median with 25–75% interquartile range, except for sex of enrolled patients. Clinical data were not available for all patients: CRP (90/92), blood leukocytes (89/92), segmented neutrophils (39/92). Mann-Whitney U test was performed to analyze statistical significance. CRP: C-reactive protein.

**Table 2 cells-12-00959-t002:** List of pathogens detected in the CAPNETZ cohort.

Pathogen(s)	*n*	%
***Streptococcus pneumoniae*** **only**	**69**	**75%**
***Streptococcus pneumoniae*** **and** **one additional pathogen**	**15**	**16.30%**
**Bacteria**	**9**	**9.78%**
*Enterobacter spp.*	1	1.09%
*Haemophilus influenzae*	2	2.17%
*Haemophilus parainfluenzae*	2	2.17%
*Legionella spp.*	1	1.09%
*Pseudomonas spp.*	1	1.09%
*Serratia liquefaciens*	1	1.09%
*Staphylococcus aureus*	1	1.09%
**Viruses**	**4**	**4.35%**
Rhinovirus	2	2.17%
Influenzavirus A/H3N2	1	1.09%
Parainfluenzavirus 4	1	1.09%
**Fungi**	**2**	**2.17%**
*Candida albicans*	1	1.09%
*Candida spp.*	1	1.09%
***Streptococcus pneumoniae*** **and** **more than one additional pathogen**	**8**	**8.70%**
**Bacterium/fungus**	**3**	**3.26%**
*Staphylococcus aureus/Klebsiella oxytoca/Candida albicans*	1	1.09%
*Pseudomonas aeruginosa/Serratia marcescens/Escherichia coli/Candida albicans*	1	1.09%
*Citrobacter* spp*/Candida albicans*	1	1.09%
**Bacterium/bacterium**	**1**	**1.09%**
*Staphylococcus aureus/Staphylococcus epidermidis*	1	1.09%
**Bacterium/virus**	**2**	**2.17%**
*Staphylococcus aureus/*coagulase-negative *staphylococcus/*RSV	1	1.09%
*Haemophilus influenzae/*Coronavirus NL 64	1	1.09%
**Virus/fungus**	**1**	**1.09%**
Parainfluenzavirus 3*/Candida spp.*	1	1.09%
**Virus/fungus**	**1**	**1.09%**
*Aspergillus fumigatus/Candida albicans*	1	1.09%
**ALL CASES**	**92**	**100%**

All CAP patients from the CAPNETZ cohort (*n* = 92, 100%) had confirmed presence of *Streptococcus pneumoniae*, 69 patients (75%) had confirmed presence of *Streptococcus pneumoniae* as the sole pathogen, 15 patients (16.30%) had confirmed presence of *Streptococcus pneumoniae* and one additional pathogen, 8 patients (8.70%) had confirmed presence of *Streptococcus pneumoniae* and more than one additional pathogen.

## Data Availability

Single-cell RNA sequencing data is made publicly available through Zenodo via https://doi.org/10.5281/zenodo.6704334 (accessed on 15 March 2023). Codes used to analyze transcriptomics data is available through GitHub at https://github.com/GenStatLeipzig/Regulation-of-acute-lung-inflammation-by-microRNA-223 (accessed on 15 March 2023).

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
