# Peer review of "MicroRNA-223 Dampens Pulmonary Inflammation during Pneumococcal Pneumonia"

_cells, 2023, doi:10.3390/cells12060959_

Round 1

Reviewer 1 Report

In this work entitled "MicroRNA-223 Dampens Pulmonary Inflammation During Pneumococcal Pneumonia". Goekeri et. al. investigate the role of miR-223 in the context of lung inflammation in a murine model of pneumococcal pneumonia and make an association of serum miR-223 levels with disease severity in pneumococcal pneumonia patients. The role of miR-223 in the resolution of inflammation has been extensively investigated in a context of sterile inflammation but its role during bacterial infections are still elusive.  

The manuscript follows a clear experimental design from which the presented results can be drawn conclusively. However, there are some points that need further clarification to strengthen the scope of the manuscript.

A graphical abstract would be great for a quick grasp of the novelty added by this work.

Specific comments related to manuscript section:

Methods:

1-      What is the rationale to use female mice in this model?

2-      It is not clear the rationale for choosing the 24h and 48h timepoints to perform the experiments. It would be helpful if the authors provide a dataset (or cite some previous work) showing the time-course of the disease in this particular model.

Results:

1-      miR-223 Is Reduced in Human Serum During Bacterial Pneumonia and Transiently 306 Increases in Murine Lungs Upon Pneumococcal Infection:

Was the concentration of total RNA elevated in the serum of the CAP patients compared to the healthy donors? Maybe there is an increase in the total mRNA in the serum during the inflammatory response and this is what is causing an apparent reduction in the mRNA for miR-223.

In mice, however, they show that there is an increase in the expression of miR-223 in the lungs of infected mice. Why is there this reduction in miR-223 in humans while it increases in mice? Is it related to the different compartment in which it was evaluated? The authors should provide the analysis of miR-223 levels in mouse serum as well for more translatable data.

2-   miR-223 Regulates Pulmonary Neutrophil Migration, Persistence and Apoptosis

There is no difference in the total number of neutrophils recovered in BALF samples from WT and miR-223 -/- mice. The difference appears at 48 hpi when taken in consideration the percentage of PMN. Is there a reduction in other leukocytes in the airways? Were only PMNs analyzed? What about macrophages, monocytes etc.?

In figure 3 it is shown that in the lungs and BALF of miR-223 -/- mice, there are more neutrophils expressing CD11b, which is commonly used as a marker of neutrophil activation. Can the authors discuss this difference?

3-      miR-223 Regulates Genes Involved in Inflammation, Granulocyte Maturation and 414 Antibacterial Defense

KO mice have more neutrophils in the lungs but there is no difference in the CFU. Why are these neutrophils not killing the bacteria?

Author Response

Dear Reviewer 1,

Please find our response in the word document in the attachment.

Reviewer 2 Report

The manuscript from Goekeri et. al presents evidence that miR-223 regulates lung inflammation during pneumococcal pneumonia in mice. In addition, levels of miR-223 were inversely associated with severity of CAP in humans. Among the mechanisms associated to the increased inflammation observed in miR-223 KO mice, the authors suggest the presence of a hyperactivated neutrophil phenotype. The study is overall clearly conducted and aids to the literature related to the role of miR-223 in controlling inflammation during disease. The authors should clarify some aspects of the study.

1)      Ethics approval project number should be included for both animal and human work. This must be included in the methods section.

2)      The authors should include a list of the pathogens associated to the CAP patients’ samples used.

3)      Macrophages are essential to control S. pneumoniae in the lungs and are important orchestrators of inflammation/resolution of inflammation.  How is the expression of miR-223 in macrophages post-infection? Are the number of macrophages changed in KO vs WT (figure 3 suggests otherwise)? Can the excess of neutrophils be related to a decreased efferocytosis ability of macrophages?   

4)      Does the absence of miR-223 in neutrophils change these cells functionally as well? (i.e. ROS, NETs, cytokine secretion, migration,etc) Does it influence the killing of bacteria?

5)      Minor: lines 303-305 seem unnecessary.

Author Response

Dear Reviewer 2,

Please find our response in the word document in the attachment.

Round 2

Reviewer 1 Report

N.A.

Reviewer 2 Report

The authors have answered all the relevant points.